# Diffusion MRS tracks distinct trajectories of neuronal development in the cerebellum and thalamus of rat neonates

Clémence Ligneul[1]*, Lily Qiu[1], William T Clarke[1], Saad Jbabdi[1], Marco Palombo[2,3†], Jason P Lerch[1,4,5†]

[1]Wellcome Centre for Integrative Neuroimaging, FMRIB, Nuffield Department of Clinical Neurosciences, University of Oxford, Oxford, United Kingdom; [2]Cardiff University Brain Research Imaging Centre (CUBRIC), School of Psychology, Cardiff University, Cardiff, United Kingdom; [3]School of Computer Science and Informatics, Cardiff University, Cardiff, United Kingdom; [4]Mouse Imaging Centre, The Hospital for Sick Children, Toronto, Canada; [5]Department of Medical Biophysics, University of Toronto, Toronto, Canada

*For correspondence:
clemence.ligneul@gmail.com

†These authors contributed
equally to this work

Competing interest: See page
16

Reviewing Editor: Susie Y
Huang, Massachusetts General
Hospital, United States

## eLife Assessment

This study presents a **valuable** investigation into cell-specific microstructural development in the neonatal rat brain using diffusion-weighted magnetic resonance spectroscopy. The evidence supporting the core claims is **solid**, with innovative in vivo data acquisition and modeling, noting residual caveats with regard to the limitations of diffusion-weighted magnetic resonance spectroscopy for strict validation of cell-type-specific metabolite compartmentation. In addition, the study provides community resources that will benefit researchers in this field. The work will be of interest to researchers studying brain development and biophysical imaging methods.

**Abstract** It is currently impossible to non-invasively assess cerebellar cell structure during early development. Here, we propose a novel approach to non-invasively and longitudinally track cell-specific development using diffusion-weighted magnetic resonance spectroscopy (MRS) in combination with microstructural modelling. Tracking metabolite diffusion allows us to probe cell-specific developmental trajectories in the cerebellum and thalamus of healthy rat neonates from postnatal day (P) 5 to P30. Additionally, by comparing different analytical and biophysical microstructural models, we can follow the differential contribution of cell bodies and neurites during development. The thalamus serves as a control region to assess the sensitivity of our method to microstructural differences between the regions. We found significant differences between cerebellar and thalamic metabolites' diffusion properties. For most metabolites, the signal attenuation is stronger in the thalamus, suggesting less restricted diffusion compared to the cerebellum. There is also a trend for lower signal attenuation and lower apparent diffusion coefficients (ADCs) with increasing age, suggesting increasing restriction of metabolite diffusion. This is particularly striking for taurine in the thalamus. We use biophysical modelling to interpret these differences. We report a decreased sphere fraction (or an increased neurite fraction) with age for taurine and total creatine in the cerebellum, marking dendritic growth. Surprisingly, we also report a U-shape trend for segment length (the distance between two embranchments in a dendritic tree) in the cerebellum, agreeing with age-matching morphometry of openly available 3D-Purkinje reconstructions. Results demonstrate that diffusion-weighted MRS probes early cerebellar neuronal development non-invasively.

# Introduction

The cerebellum is a well-conserved structure across species and particularly across mammals. The cerebellar cortex is a highly folded structure organised in three layers: the molecular layer (containing the notable Purkinje cell dendritic trees), the Purkinje layer (containing the somas of Purkinje cells), and the granular layer (containing the granule cells). Projections to the deep cerebellar nuclei form the cerebellar white matter. Despite its small size (compared to the cerebrum), it contains more than half of the total brain neurons: this is mainly due to the granule cells, very small and densely packed cells, some of the smallest of the brain, presenting a very simple morphology (*Haines and Dietrichs, 2012*). On the other hand, the Purkinje cells are one of the largest types of neurons and most complex cells in the brain. Cerebellar development is protracted (*Szulc et al., 2015*), starting from birth (or the last trimester of pregnancy) to a few weeks old in rodents (or 2 y.o. in humans) (*Sathyanesan et al., 2019*). The cerebellum is involved in many basic and high-level functions, from motor control to social skills, and it is known to be affected in a number of neurodevelopmental disorders (*Stoodley, 2016*).

During early development, neurons arborise. Their morphology becomes more complex and exhibits denser branching. In rodents, Purkinje cells' dendritic trees start growing at birth. In 30 days, they evolve from a simple soma (~10 μm diameter) to a large cell with a highly branched and planar dendritic tree (soma diameter ~20 μm, area ~10,000 μm$^2$ in the mouse; *Beekhof et al., 2021*). Purkinje cell morphology can be altered in autism spectrum disorders (ASD), as shown post-mortem, in humans (*Fatemi et al., 2002*) and multiple mouse models of ASD. Specifically, they show an increased spine density in Tsc1 mutant mice (*Tsai et al., 2012*), an overgrowth of the dendritic tree and abnormal branching in PTEN KO mice (*Cupolillo et al., 2016*), a reduction of the dendritic tree in AUTS2 mice (*Yamashiro et al., 2020*), and general alterations in developmental disorders (*Kim et al., 2011*; *Robinson et al., 2012*; *Usui et al., 2017*; *Sundberg et al., 2018*).

All cerebellar outputs transit via the deep cerebellar nuclei, and a couple of them project to the thalamus before spreading to the cortex. The thalamus is considered a relay station in the brain and contains about 60 nuclei on unique pathways. Some thalamic nuclei are involved in psychiatric disorders, and more specifically, the microstructure of the mediodorsal nucleus and the pulvinar nucleus is altered in schizophrenia and psychosis (*Hwang et al., 2022*). Some other nuclei are involved in neurodevelopmental disorders (e.g. the lateral reticular nuclei in ASD; *Krol et al., 2018*). Generally, these disorders are associated with an altered thalamic connectivity (*Hwang et al., 2022*). Thalamocortical projections are crucial for the cytoarchitectural development of the cortex (*Antón-Bolaños et al., 2018*; *Sato et al., 2022*).

Neurodevelopmental disorders, or even psychiatric disorders with a neurodevelopmental origin, such as schizophrenia (*Fatemi and Folsom, 2009*), generally need better neurobiological characterisation during development (*Ismail and Shapiro, 2019*).

Rodents reach adulthood after a couple of months and are an ideal model for mammal brain development. While brain cell developmental trajectories are key to assessing neurodevelopment, there is a lack of translational techniques allowing for a non-invasive measure of their properties. Typically, EEG and MEG signals contain brain activity cell-specific information. Still, it is hard to retrieve in practice, and the existing forward or inverse models notably use prior information on cell morphology. PET can be cell-specific and provide information about metabolism, but it is blind to cell morphology and relies on radioactive ligands, making it very expensive and potentially harmful. fNIRS and ultrasounds are too indirect to measure cell-specific information as they rely on the haemodynamic response. Last but not least, MRI is non-invasive and non-harmful. The MR signal can be sensitised to assess different tissue properties. In the following paragraphs, we will summarise the importance and limitations of some MR methods sensitive to brain structure and microstructure during development and introduce our primary method, namely diffusion-weighted MR spectroscopy (dMRS).

(*Structural MRI: anatomy*) Manganese-enhanced MRI (MEMRI) is particularly suited to studying whole-brain early development in rodents. It provides high structural contrast and a fast acquisition time in neonates (*Szulc et al., 2015*; *Qiu et al., 2018*). Still, the method is blind to the microstructural changes that lead to the volumetric changes reported.

(*dMRI: microstructure*) Diffusion-weighted MRI (dMRI) measures the diffusion properties of water in the brain: restricted and hindered by cell membranes, water diffusion provides an indirect measure of the tissue microstructure (*Alexander et al., 2019*). dMRI is versatile and applied in numerous applications, including the characterisation of the microstructural organisation during neurodevelopment

in rodents and, to a lesser extent, in humans. Comparison with post-mortem microscopy shows a correlation between a high cortical fractional anisotropy (FA) at birth and the presence of radial glia. The latter progressively disappears in the first week along with an FA decrease, illustrating the potential of dMRI to characterise neurodevelopment (*Hüppi, 2010*; *Chokshi et al., 2011*). In the cerebellum, the dense folding limits diffusion imaging to the exploration of white matter or ex vivo samples (*Zheng et al., 2023*).

(*dMRS: cell-specific microstructure*) Water is ubiquitous and can only partially disentangle the contribution from different tissue compartments (e.g. intra- and extracellular space). dMRS measures the diffusion properties of metabolites. Brain metabolites are mostly intracellular, with some more specific to neurons (e.g. *N*-acetyl-aspartate [NAA] and glutamate [Glu]) and others more specific to glial cells (e.g. choline compounds, tCho, and myo-inositol, Ins); therefore, different cellular compartments can be resolved (*Ronen and Valette, 2015*). However, metabolites are $10^4$-fold less concentrated than water, requiring large measurement volumes for dMRS (typically ~50–100 mm$^3$ in rodents), and long acquisition times. As for dMRI, some dMRS acquisition parameters, such as the diffusion weighting and diffusion time, enable the signal to be sensitised to different cellular morphological scales (e.g. cellular processes, cell bodies, cell extension). Using multiple diffusion times and a post hoc model, cellular morphological parameters can be estimated from dMRS (*Ligneul et al., 2024*). They correlate with neuronal and glial morphologies extracted from histological measures in the healthy, adult mouse (*Palombo et al., 2016*; *Palombo et al., 2017*). More specifically, it has been shown that myo-inositol diffusion properties reflect astrocytic morphology in an induced model of astrocytic hypertrophy (*Ligneul et al., 2019*) and in a mouse model of demyelination where astrocytic activation was expected (*Genovese et al., 2021*). Choline compound diffusion properties were associated with microglial activation in human disease (*Ercan et al., 2016*) or in an LPS-induced brain inflammation model (*de Marco et al., 2022*). However, to our knowledge, very few studies have used dMRS to probe changes in neuronal morphology, and none have attempted to probe neuronal development in early life.

Here, we propose a non-invasive method to monitor cellular brain development. We focus on two regions with different developmental timelines: the cerebellum and the thalamus. As described above, the cerebellum expands greatly after birth, when the Purkinje cells' dendritic trees start growing. This slow maturation exposes the cerebellum longer to developmental insults. In contrast to the cerebellum, the thalamus develops relatively quickly, and its neurons expand and complexify more rapidly (*Szulc et al., 2015*; *Takeuchi et al., 2014*; *Figure 1A*).

By contrasting these two regions, we want to assess the sensitivity of the proposed method. We scanned developing rats from postnatal day 5 (P5) to P30. We first used MEMRI to confirm the protracted cerebellar development compared to the thalamus (*Figure 1B*). We then acquired diffusion properties of metabolites to probe short- and long-range restriction (i.e. soma restriction and branching). To interpret our data, we applied two biophysical models (one analytical and one computational) to characterise the specific changes in microstructure (*Figure 1C*). We expected monotonic changes in microstructural parameters in both regions, reflecting cell growth, with thalamic properties plateauing earlier than cerebellar properties. This is mostly what we report in our observations. However, we report a U-shape trend for the cerebellar segment length (L$_{segment}$, the distance between two embranchments of a dendritic tree) that matches openly available 3D cell reconstructions. In contrast to common findings in the adult brain, we report that tNAA and Glu do not provide reliable morphological neuronal markers in neonates. We identify instead total creatine (tCr) and taurine (Tau) as more reliable markers for tracking early neuronal development. We believe that this is a promising method to track early neuronal development, particularly in the cerebellum, where microstructure changes drastically postnatally.

## Results
### Structural imaging confirms protracted cerebellar growth

Representative slices from intra-age non-linear registration are shown in *Figure 2A*, underlying the cerebellar unfolding. Both cerebellar and thalamic volumes are just below 50 µl at P5 (*Figure 2B*). However, the segmentation at P5 is poor, as explained in Methods (*Figure 2—figure supplement 1*);

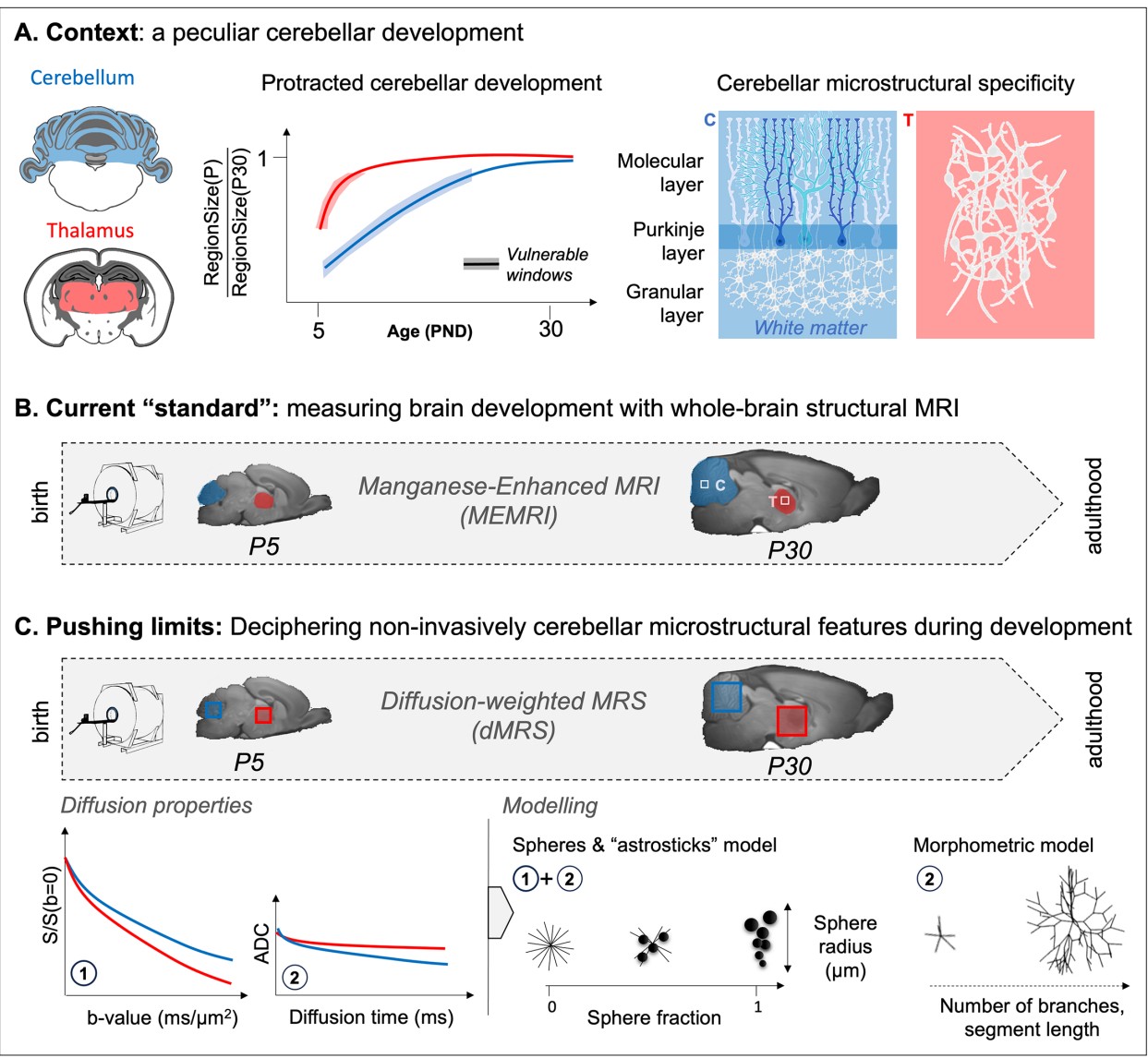

**Figure 1.** Overview of the rationale of the study. (**A**) Context: the cerebellum exhibits protracted growth compared to other brain regions (e.g. thalamus), making it a particularly vulnerable region for neurodevelopmental disorders. Its specific microstructural organisation (i.e. layered with Purkinje cells and granular cells) changes between birth and adulthood. (**B**) The current 'standard' is to probe regional growth with manganese-enhanced MRI (MEMRI). However, it is not sensitive to the underlying microstructural changes. (**C**) Pushing limits: diffusion-weighted MRS (dMRS) can potentially assess cell-specific microstructural changes in given regions during development. Different measures of metabolites' diffusion properties (apparent diffusion coefficient [ADC] at long diffusion times, signal attenuation [S] at high diffusion weighting, b-values) can be interpreted with biophysical modelling and help characterise the nature of the microstructural changes.

hence, data at this time point should be treated cautiously, although the cerebellum and the thalamus, which are medial line structures, are less affected by the atlas distortions.

The cerebellum triples in volume, between P5 and P30, reaching 90% of its P30 volume at P20, while the thalamus reaches 90% of the P30 volume at P15.

## Metabolic profile changes with age

The metabolic profile changes noticeably between P5 and P30 as shown in *Figure 3A*. The absolute concentrations of metabolites, estimated by LCModel using the $b_0$ averaged spectra, shown in *Figure 3A*, were normalised by the water content at $b_0$ (*Figure 3B*). The metabolite concentrations reported were neither corrected for T1 (TR = 2.2 s) nor for the non-null diffusion weighting and are therefore only apparent concentrations. Conventionally, metabolite absolute concentrations

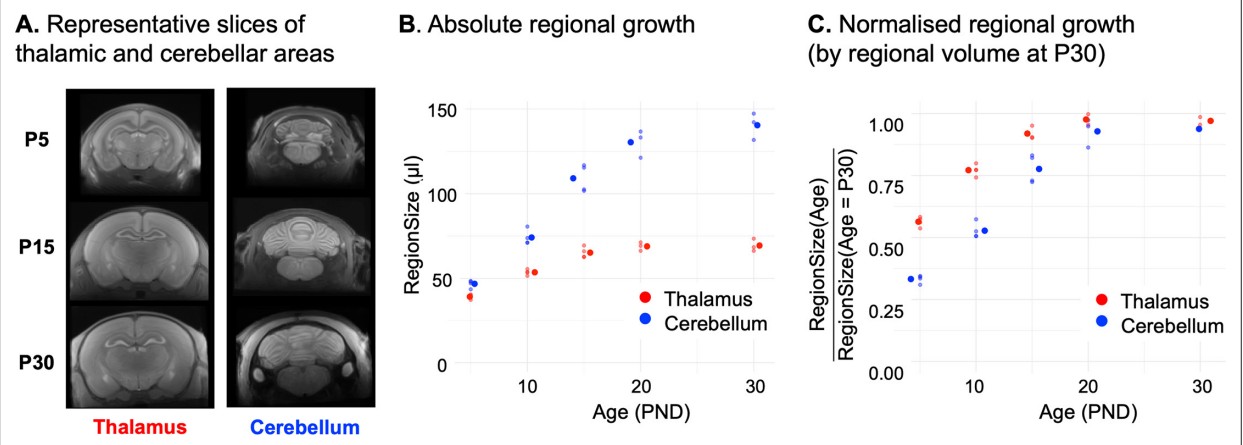

**Figure 2.** Manganese-enhanced MRI confirms protracted cerebellar growth in rat pups. (**A**) Representative slices around the thalamus and cerebellum of non-linearly averaged images acquired at postnatal day 5 (P5), P15, and P30. (**B**) Absolute volumetric changes are shown for the segmented thalamus and cerebellum. Large dots correspond to the mean and small dots to individual data points. (**C**) Normalised volumetric changes. Volume at each age is normalised by the P30 volume. Large dots correspond to the mean and small dots to individual data points.

The online version of this article includes the following figure supplement(s) for figure 2:

**Figure supplement 1.** Representative segmented atlases at each time point.

are normalised by the water content in the voxel or by the tCr absolute concentration, which are considered stable in the healthy brain. However, this study compares regions with different relaxation properties at different ages, with different metabolite profiles in adulthood. The water relaxation properties are different between the thalamus and cerebellum, and both water relaxation properties and content change with age (**Kumar et al., 2011**, **Tkác et al., 2003**). The tCr concentration changes substantially with age (about +50% in the cerebrum between P7 and P28) and is known to be higher in the cerebellum in adulthood (about +40–50% compared to the cerebrum) (**Tkác et al., 2003**; **Tkáč et al., 2004**). As an alternative to normalisation by water content, we propose in **Figure 3—figure supplement 1** a normalisation by the absolute MM concentration provided by LCModel, confirming trends reported by water normalisation. MM resonances mainly come from the cytosolic proteins; they are a direct probe of the tissue and do not depend on the voxel CSF content. Although their content might slightly vary between regions in the healthy brain, they make a reasonable intracellular reference if SNR allows. Between P5 and P30, apparent metabolite concentrations increase for tNAA, Glu, Gln, and Ins, and decrease for PE and Tau (only in the thalamus). tCho and Gly exhibit slightly different trends depending on the normalisation mode (**Figure 3—figure supplement 1**). MM and water content changes with age are reported in **Figure 3—figure supplement 2**. Note that the increase in tNAA concentration with age is protracted in the cerebellum compared to thalamus and has not reached a plateau at P30. These results generally match the metabolic concentrations measured with gold-standard classical MRS in the striatum, cortex, and hippocampus of developing rats reported in **Tkác et al., 2003**.

## Metabolite diffusion properties differ between regions and change with age

Thalamic (**Figure 4A**) and cerebellar (**Figure 4B**) signal attenuations up to b=30 ms/μm² and apparent diffusion coefficients (ADCs) at TM = 100–1000 ms are shown for tNAA, Glu, tCr, Tau, tCho, and Ins. These diffusion properties differ between both regions at each age (see **Figure 4—figure supplement 1** for a region-to-region visual comparison). For most metabolites (except tCho), the signal attenuation is stronger in the thalamus, suggesting less restricted diffusion compared to the cerebellum. There is also a trend for lower signal attenuation and lower ADCs with increasing age for some metabolites, suggesting increasing restriction of metabolite diffusion. This is particularly striking for Tau in the thalamus. To give an estimation of the MRS fit performance, CRLBs are given for tNAA, Glu, tCr, Tau, tCho, and Ins for the different diffusion conditions in **Supplementary file 1**.

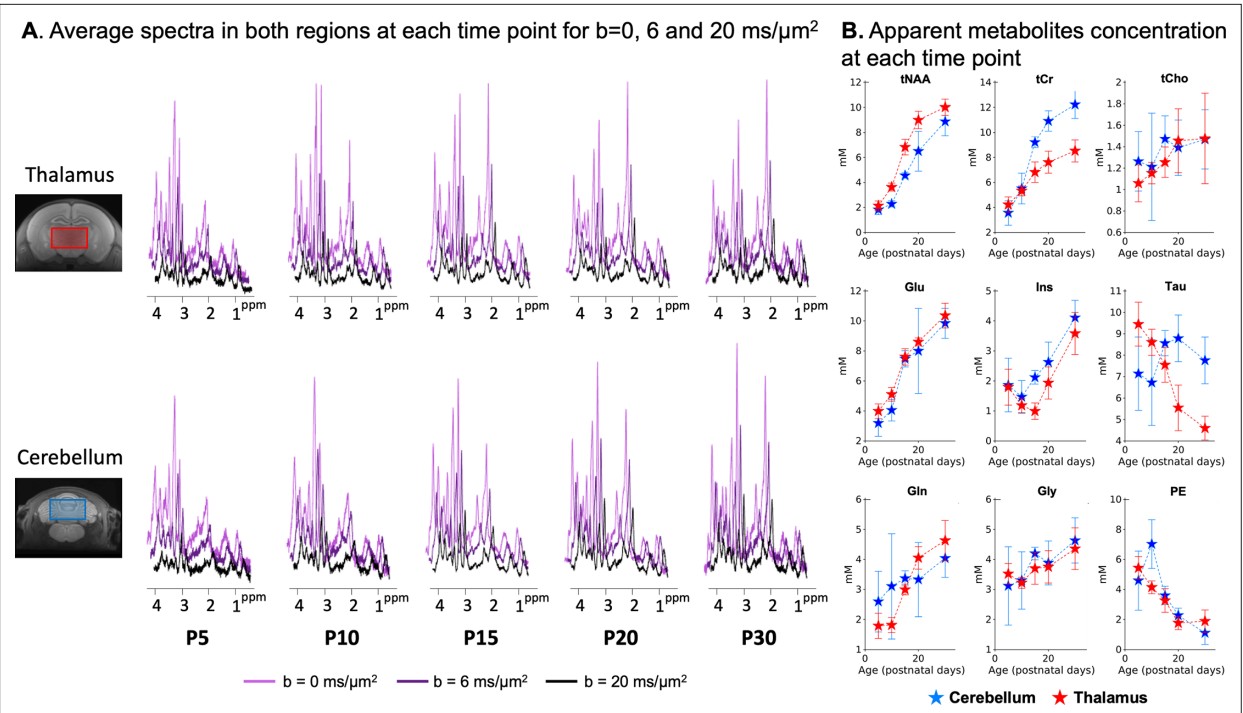

**Figure 3.** Metabolic profile changes postnatally. (**A**) Spectra were binned by time point, region, and b-value, and averaged after phase and frequency correction. The SNR is clearly lower at postnatal day 5 (P5)/P10 in both regions: the voxel is about half of the P30 voxel size, and more animals were outliers. The SNR is also lower at P30: the rat size was slightly too big for the mouse cryoprobe at this age, so the skull is not at the top of the probe and the sensitivity is suboptimal. (**B**) Individual spectra at b=0.035 ms/µm² were quantified with LCModel, and absolute metabolite concentrations were normalised by the water content and levelled to get Signal(tCr,P30)/Signal(water,P30)~8 in the thalamus. See *Figure 3—figure supplement 1* for a comparison with normalisation by MM.

The online version of this article includes the following figure supplement(s) for figure 3:

**Figure supplement 1.** Cerebellar (blue) and thalamic (red) metabolic changes with age.

**Figure supplement 2.** Cerebellar (blue) and thalamic (red) molecular and water content changes with age.

## Biophysical modelling underlines different developmental trajectories of cell microstructure between the cerebellum and the thalamus

Data were fitted with a spheres + 'astrosticks' model and a morphometric model (more details in the Methods section; for model comparison, see *Figure 4—figure supplement 2*). The spheres + 'astrosticks' model captures diffusion within spheres (representing cell bodies) and randomly oriented sticks (representing cell processes/neurites): parameters extracted are sphere fraction, $f_{sphere}$ (or equivalently neurite fraction as $1-f_{sphere}$) and cell radius, R. The parameter space (sphere radius vs. sphere fraction) is represented in the third column of *Figure 4A* (cerebellum) and *Figure 4B* (thalamus) for tNAA, Glu, tCr, Tau, tCho, and Ins at all ages (see *Figure 4—figure supplement 3* for a region-to-region visual comparison).

Mean values and standard deviations are reported in *Table 1*, along with the results from the linear mixed effect model ~Age + Region + Age:Region + (1|PupID). The fits on averaged data, for binned spectra by age and region, are shown in *Figure 4—figure supplement 4*.

The morphometric model, only applied to averaged ADCs at long $t_d$, interprets diffusion in branched structures: the segment length, $L_{segment}$, is reported on the last column of *Figure 4A and B* (see *Figure 4—figure supplement 5* for a region-to-region visual comparison and *Figure 4—figure supplement 6* for all parameters estimated by the morphometric model: $D_{intra}$, $N_{Branch}$, $L_{segment}$).

### Neuronal markers

tNAA and Glu are primarily concentrated in neurons in adulthood.

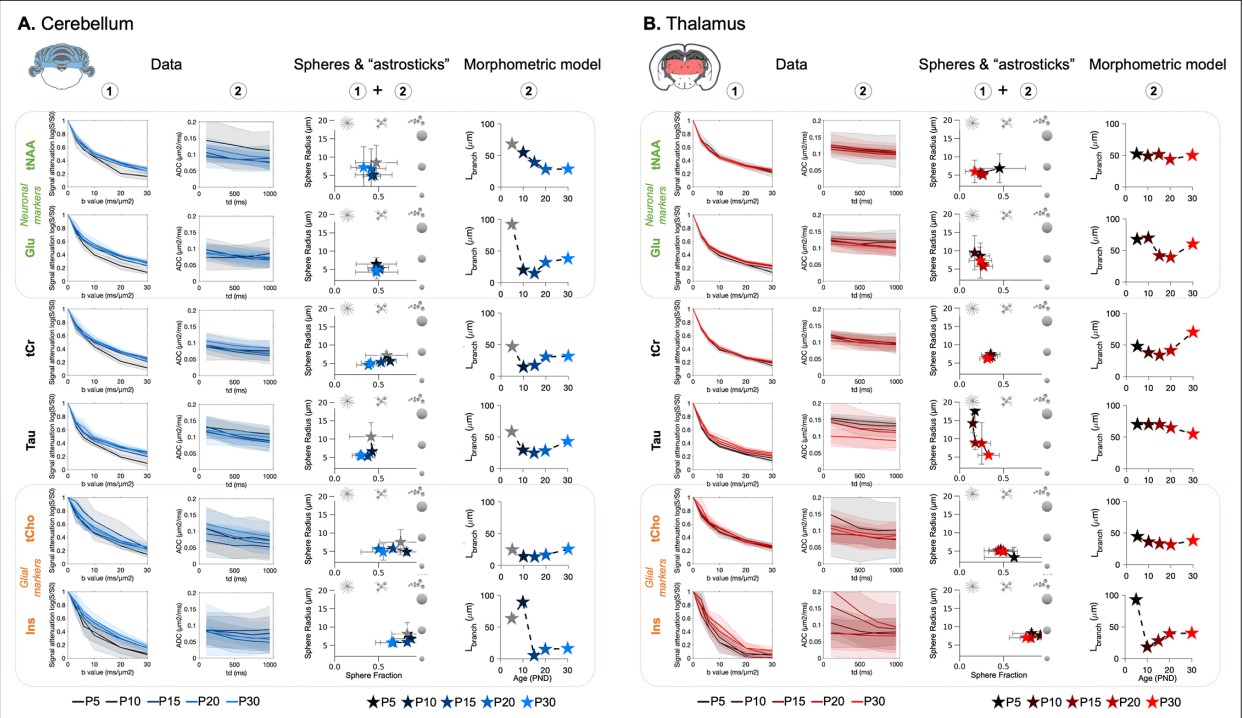

**Figure 4.** Biophysical modelling issued from metabolites diffusion properties shows differential developmental trajectories in the cerebellum (**A**) and in the thalamus (**B**). The first column (**A, B**) reports the signal attenuation as a function of b-value at TM = 100 ms, and the second column reports the apparent diffusion coefficient (ADC) varying with diffusion time. Shadowed error bars represent the standard deviations. Modelling results from the spheres+'astrosticks' model are reported in the third column (**A, B**). Postnatal day 5 (P5) is shaded in the cerebellum due to possible motion artefact. The parameter space (sphere fraction, sphere radius) is displayed. Error bars represent the standard deviations. $L_{length}$, modelling output from the morphometric model, is shown in the fourth column (**A, B**). P5 is shaded in the cerebellum due to possible motion artefact.

The online version of this article includes the following figure supplement(s) for figure 4:

**Figure supplement 1.** Visual comparison of diffusion properties between regions and at each time point.

**Figure supplement 2.** Analytical model residuals.

**Figure supplement 3.** Visual comparison of the parameter space (sphere fraction, sphere radius) between the cerebellum and the thalamus at each time point for the 'astrosticks + spheres' model ($D_{intra} = 0.5$ μm²/ms).

**Figure supplement 4.** Parameter space (sphere fraction, sphere radius) between the cerebellum and the thalamus at each time point for the 'astrosticks + spheres' model ($D_{intra} = 0.5$ μm²/ms) on averaged neonates.

**Figure supplement 5.** Visual regional comparison of parameters estimated from the morphometric model.

**Figure supplement 6.** Estimations from the morphometric model of $D_{intra}$, $N_{branch}$, $L_{segment}$ are shown for the six metabolites in two regions.

**Figure supplement 7.** DTI results in the cerebellum and thalamus.

**Figure supplement 8.** Parameter space (sphere fraction, sphere radius) between the cerebellum and the thalamus at each time point for the 'astrosticks + spheres' model ($D_{intra} = 0.7$ μm²/ms).

**Figure supplement 9.** Average macromolecule signal attenuation in both regions at each time point coming from the LCModel fit.

- Thalamus: The sphere fraction and the radius estimated from tNAA and Glu diffusion properties do not change much with age. For both metabolites, $f_{sphere}$ is around 0.25 and R decreases to 5–6 μm. $L_{segment}$ is stable with age for tNAA, but shows a reduction at P15-P20 for Glu.
- Cerebellum: The sphere fraction and the radius estimated from tNAA diffusion properties vary with age. $f_{sphere}$ nears 0.5 in the cerebellum and R decreases below 5 μm for tNAA and Glu. $L_{segment}$ decreases until P15 for both tNAA and Glu.

Between regions, sphere fraction differs significantly (**Table 1**).

## Glial markers

tCho and Ins are primarily concentrated in glial cells in adulthood. The general trend is a high sphere fraction (~0.7–0.9) for Ins and moderately high (~0.5–0.6) for tCho. The radius estimated from tCho

**Table 1.** Mean values and standard deviations for parameters extracted from the spheres + 'astrosticks' model (sphere radius and sphere fraction).

Statistics come from a linear model accounting for age and region as fixed effects. The interaction term was included only if it improved the fit significantly (c.f. Methods). p-Values highlighted in yellow pass the α=0.05 threshold after multiple comparison correction and green p-values pass the α=0.01 threshold (Bonferroni correction, 12 hypotheses).

| | Cerebellum | | | | | Thalamus | | | | | (Radius, Fraction)~Region*Age + (1\|PupID) | | |
|---|---|---|---|---|---|---|---|---|---|---|---|---|---|
| | | | | | | | | | | | *p-Values before multiple comparison correction* | | |
| Age (PND) | 5 | 10 | 15 | 20 | 30 | 5 | 10 | 15 | 20 | 30 | Region | Age | Region:Age |
| | NAA + NAAG | | | | | NAA + NAAG | | | | | | | |
| Radius (μm) | 8.51 | 5.18 | 4.78 | 6.92 | 7.21 | 6.84 | 5.52 | 5.05 | 5.24 | 5.97 | 3.69E-01 | 7.13E-01 | |
| std(Radius) (μm) | 4.72 | 1.10 | 0.58 | 5.34 | 5.66 | 3.96 | 0.77 | 0.59 | 0.55 | 3.08 | | | |
| $f_{sphere}$ | 0.47 | 0.45 | 0.42 | 0.42 | 0.33 | 0.46 | 0.24 | 0.27 | 0.26 | 0.17 | 1.20E-04 | 9.11E-04 | |
| std($f_{sphere}$) | 0.24 | 0.21 | 0.10 | 0.17 | 0.15 | 0.30 | 0.08 | 0.04 | 0.03 | 0.11 | | | |
| | Glu | | | | | Glu | | | | | | | |
| Radius (μm) | 6.59 | 5.49 | 5.04 | 4.26 | 4.79 | 8.57 | 9.46 | 6.26 | 5.71 | 7.32 | 1.14E-03 | 1.74E-02 | |
| std(Radius) (μm) | 1.78 | 0.91 | 0.24 | 1.88 | 0.12 | 2.42 | 4.70 | 0.62 | 0.62 | 4.82 | | | |
| $f_{sphere}$ | 0.47 | 0.50 | 0.51 | 0.48 | 0.47 | 0.23 | 0.17 | 0.27 | 0.28 | 0.24 | 2.27E-06 | 7.45E-01 | |
| std($f_{sphere}$) | 0.24 | 0.12 | 0.04 | 0.24 | 0.09 | 0.11 | 0.06 | 0.07 | 0.10 | 0.13 | | | |
| | Cr + PCr | | | | | Cr + PCr | | | | | | | |
| Radius (μm) | 7.17 | 5.61 | 5.31 | 4.54 | 5.07 | 7.38 | 6.87 | 6.53 | 6.45 | 6.18 | 1.49E-01 | 1.83E-03 | 5.13E-02 |
| std(Radius) (μm) | 1.09 | 0.58 | 0.32 | 0.78 | 0.50 | 0.92 | 0.57 | 0.31 | 0.69 | 0.67 | | | |
| $f_{sphere}$ | 0.59 | 0.63 | 0.53 | 0.39 | 0.41 | 0.35 | 0.36 | 0.34 | 0.31 | 0.32 | 1.32E-07 | 3.26E-01 | 6.55E-03 |
| std($f_{sphere}$) | 0.24 | 0.07 | 0.05 | 0.14 | 0.10 | 0.11 | 0.07 | 0.02 | 0.04 | 0.09 | | | |
| | Tau | | | | | Tau | | | | | | | |
| Radius (μm) | 10.63 | 6.63 | 5.32 | 5.41 | 5.56 | 17.57 | 14.18 | 8.90 | 8.73 | 5.50 | 4.27E-08 | 3.53E-12 | 7.41E-04 |
| std(Radius) (μm) | 3.83 | 1.76 | 0.83 | 0.83 | 0.85 | 2.97 | 2.58 | 1.88 | 5.67 | 0.57 | | | |
| $f_{sphere}$ | 0.42 | 0.43 | 0.38 | 0.31 | 0.29 | 0.18 | 0.15 | 0.18 | 0.25 | 0.33 | 2.51E-10 | 2.82E-04 | 8.66E-06 |
| std($f_{sphere}$) | 0.25 | 0.05 | 0.08 | 0.06 | 0.09 | 0.03 | 0.03 | 0.05 | 0.10 | 0.12 | | | |
| | tCho | | | | | tCho | | | | | | | |
| Radius (μm) | 7.83 | 4.88 | 5.83 | 5.59 | 4.83 | 3.28 | 5.35 | 4.91 | 5.10 | 4.86 | 5.81E-04 | 2.05E-01 | 9.54E-03 |
| std(Radius) (μm) | 3.48 | 1.03 | 0.53 | 0.58 | 2.24 | 2.77 | 0.59 | 0.56 | 1.07 | 0.87 | | | |
| $f_{sphere}$ | 0.80 | 0.82 | 0.67 | 0.50 | 0.55 | 0.62 | 0.47 | 0.50 | 0.44 | 0.50 | 4.20E-04 | 5.18E-03 | |
| std($f_{sphere}$) | 0.19 | 0.20 | 0.14 | 0.07 | 0.25 | 0.34 | 0.13 | 0.11 | 0.16 | 0.16 | | | |
| | Ins | | | | | Ins | | | | | | | |
| Radius (μm) | 8.80 | 6.81 | 5.89 | 5.74 | 5.75 | 8.18 | 7.60 | | 6.93 | 7.00 | 2.19E-03 | 2.11E-03 | |
| std(Radius) (μm) | 3.11 | 0.56 | 0.51 | 0.60 | 0.75 | 1.05 | 1.16 | | 1.14 | 0.79 | | | |
| $f_{sphere}$ | 0.83 | 0.87 | 0.83 | 0.67 | 0.65 | 0.82 | 0.92 | | 0.81 | 0.76 | 2.28E-01 | 1.44E-02 | |
| std($f_{sphere}$) | 0.20 | 0.22 | 0.15 | 0.15 | 0.18 | 0.21 | 0.14 | | 0.13 | 0.19 | | | |

differs significantly between regions. Note the trend of decreasing sphere fraction in the cerebellum with age. Data at P15 for Ins in the thalamus were individually not reliable enough to apply the spheres + astrosticks model (c.f. (*Supplementary file 1*) and the particularly higher CRLBs at P15 for Ins).

## Non-specific markers

tCr and Tau are not known to be compartmentalised in a specific cell type.

- Thalamus: Parameter estimates for tCr diffusion properties do not change substantially with age ($f_{sphere} \sim 0.35$, $R \sim 6.5$ µm). Tau parameter estimation varies a lot with age: R shows a substantial decrease and $f_{sphere}$ increases.
- Cerebellum: $f_{sphere}$ decreases from 0.63 (P10) to 0.41 (P30) for tCr, but R is stable. Tau follows a similar pattern, although with an overall lower $f_{sphere}$ (0.43 at P10 to 0.29 at P30). $L_{segment}$ increases in the cerebellum from P10 for tCr and increases P15 for Tau.

Between regions, the sphere fraction differs significantly for both metabolites (*Table 1*), but only Tau exhibits a significant change with age, and this change is different between regions (significant interaction term). For Tau, the radius is significantly different between regions, changes significantly with age, and this change is different between regions (significant interaction term).

## Discussion

As in *Szulc et al., 2015*, we used MEMRI to validate the protracted development of the cerebellum in rat neonates (*Sathyanesan et al., 2019*), unfolding and tripling volume between P5 and P30. Early developmental cell-specific cerebellar and thalamic microstructural features were assessed with dMRS.

We specifically report diffusion properties for neuronal markers tNAA and Glu, glial markers tCho and Ins, and the non-specific markers tCr and Tau. Metabolites exhibit significantly different diffusion properties between regions, demonstrating the relevance of dMRS to probing early cell structure development. $f_{sphere}$ is notably higher in the cerebellum for metabolites compartmentalised fully (tNAA, Glu) or partially (tCr, Tau) in neurons, suggesting the influence of the cell-body dense granular layer of the cerebellum. $f_{sphere}$ possibly reflects the relative volume of sphere-like structures (e.g. cell bodies) compared to the relative volume of fibrous structures (e.g. cell processes, fibres).

A study of human WM using diffusion MRI reported an increase in FA and a decrease in MD with age in early life (*Hüppi, 2010*). In mice, the same relationship was found in both WM and GM (*Chahboune et al., 2009*). We report similar dMRI trends, in both cerebellum and thalamus (*Figure 4—figure supplement 7*).

We also report dMRS changes with age in both regions. The general trend observed for lower signal attenuation and ADC decrease (for some metabolites) with increasing age aligns with these reported dMRI observations and could reflect overall microstructural complexification. However, the different nature of the signal (metabolites: mostly intracellular and low compartmental exchange vs. water: ubiquitous and fast exchange) prevents close co-interpretation.

$f_{sphere}$(tCr, Tau) decreases with age in the cerebellum, meaning that tCr and Tau are located in compartments where the relative contribution of sphere-like structures (e.g. cell-bodies) decreases. In other words, it reflects cerebellar cell growth and complexification (*Figure 4A*, *Figure 5A*). In the cerebellum, while soma morphological maturation mostly happens in the first week (*McKay and Turner, 2005*), Purkinje cell dendritic expansion is rapid before P15 but then continues to mature up to the end of the 4th week (*McKay and Turner, 2005*). Tau diffusion properties might reflect the development of Purkinje cells: (i) in cats, which cannot synthesise Tau and must source it from their diet, Tau dietary deficiency delays cortical and cerebellar maturation (*Aerts and Van Assche, 2002*), meaning that Tau is also crucial in early development, and (ii) in the cerebellum, immunohistochemistry studies locate Tau in adult Purkinje cells (*Madsen et al., 1985*; *Magnusson et al., 1988*; *Hussy et al., 2000*). These observations align well too with the high Tau levels we still detect in the cerebellum at P30 (*Figure 3B*).

The diffusion properties of glial metabolites, $f_{sphere}$(Ins) and $f_{sphere}$(tCho), also tend to decrease with age. This could be explained by another cerebellar peculiarity: Bergmann glia develop long processes along Purkinje cells, and this specific growth could contribute to the $f_{sphere}$(tCr,Tau) decrease.

Validation would require an atlas of all types of cerebellar cellular processes and bodies with estimates of their relative contribution. Such a resource would be immensely valuable but does not exist. Using literature data, we use the relative thicknesses of the EGL, IGL, PL ('cell body-like' layers) and ML ('fibre-like' layer) to derive $f_{sphere,LT}$, a proxy of $f_{sphere}$, as illustrated in *Figure 5A* and reported in *Supplementary file 2* and *Figure 5—figure supplement 1*. $f_{sphere,LT}$ represents the contribution of sphere-rich layers compared to fibre-rich layers. The evolution of $f_{sphere,LT}$ trend with age is comparable to $f_{sphere}$(tCr). Although $f_{sphere}$(tCr) and $f_{sphere}$(Tau) have similar trends, $f_{sphere}$(Tau) values are overall lower and below $f_{sphere,LT}$ values (*Figure 5—figure supplement 1*). This lends weight to the idea that Tau is

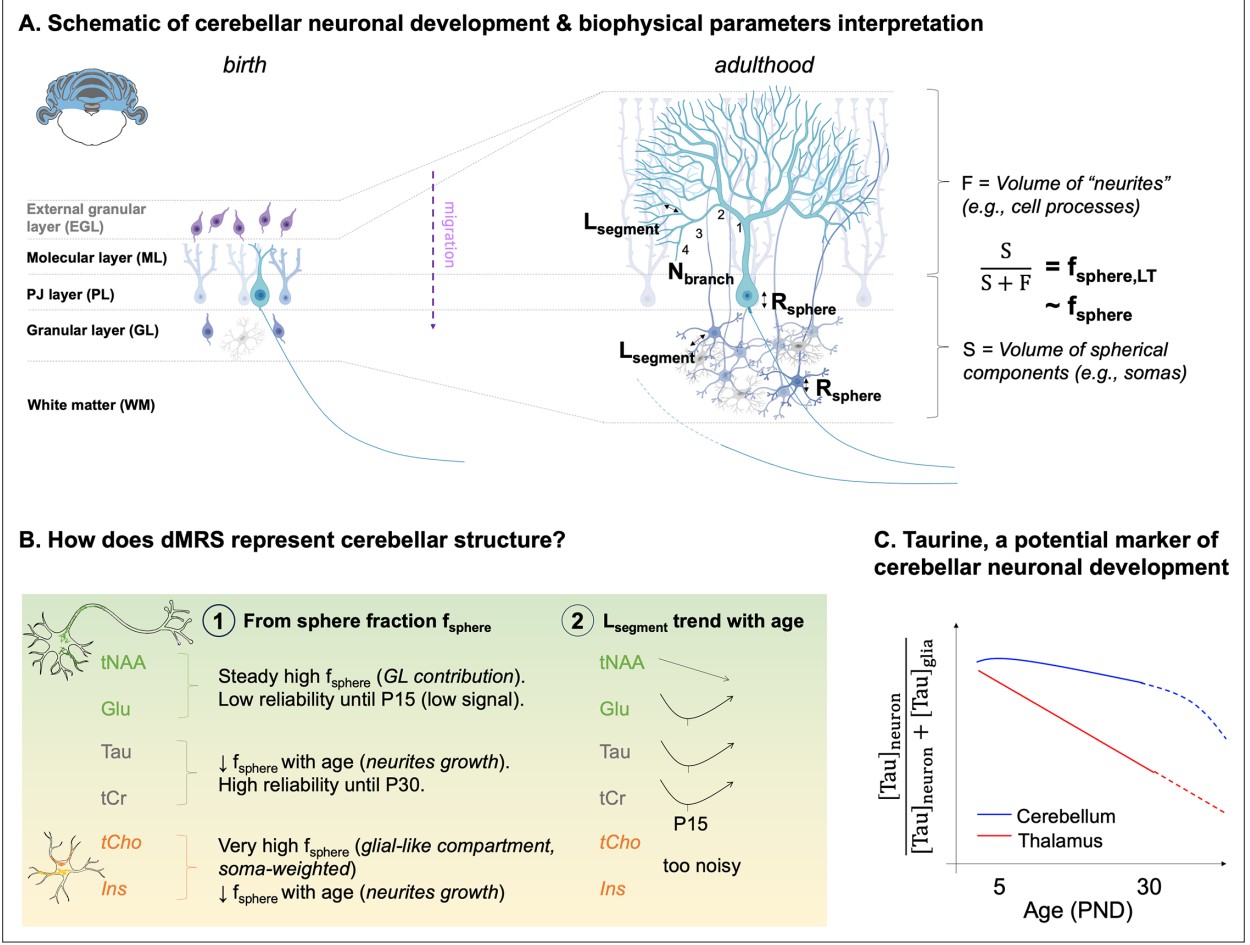

**Figure 5.** The biophysical parameters are interpreted based on the cerebellar microstructural development and main results are summarised. Italic font corresponds to possible interpretations. (**A**) Schematic of cerebellar neuronal development and biophysical modelling parameters interpretation. $L_{segment}$ is interpreted as the distance between two branches of the dendritic tree, $N_{Branch}$ as the number of embranchments. The sphere radius $R_{sphere}$ is interpreted as the average soma radius (i.e. granule and Purkinje cells). $f_{sphere,LT}$ represents the ratio of cerebellar layer thicknesses between the layers rich in sphere-like cell components (e.g. somas in the I/EGL, GL, and PL) and the layers rich in fibrous cell components (e.g. cell processes or 'neurite' in the ML). $f_{sphere,LT}$ can be seen as a rough approximation of $f_{sphere}$ and is inversely related to the dendritic tree growth, see **Supplementary file 2**. (**B**) Summary of the cerebellar results coming from the modelling. (1) and (2) refer to the models depicted in **Figure 1**. (**C**) Tau seems to change compartments in the thalamus with age, going from neuronal-like compartments (low sphere fraction) to glial-like compartments (high sphere fraction), also indicated by its important signal drop with age. It suggests that Tau is a marker of neuronal maturation. Its concentration remains high up to postnatal day 30 (P30) in the cerebellum. Tau properties in the cerebellum (decreasing sphere fraction + relatively low sphere fractions) and contrast with the thalamus suggest that Tau could be a marker of cerebellar neuronal development.

The online version of this article includes the following figure supplement(s) for figure 5:

**Figure supplement 1.** Literature estimate of $f_{sphereL,T}$ (as described in **Supplementary file 2**, pale rounds on the graph) as a function of age compared to $f_{sphere}$(tCr) (black triangles) $f_{sphere}$(Tau, tCho, Ins) (stars).

preferentially located in cerebellar neurons during cerebellar development (and potentially less so in granule cells). In comparison, tCr, which appears more evenly spread between all cerebellar cell types, tracks the evolution of $f_{sphere,LT}$ more closely. Note that Tau has a higher $D_{intra}$ (**Ligneul and Valette, 2017**) compared to the other metabolites, potentially affecting the parameter estimation given for Tau. As described in Methods, the fit for Tau was substantially improved for $D_{intra} > 0.6 \ \mu m^2/ms$. The results for $D_{intra} = 0.7 \ \mu m^2/ms$ are similar for all the other metabolites (**Figure 4—figure supplement 8**), and generally improved the estimated sphere radius standard deviation. Additionally, the evolution with age of $(R, f_{sphere})$ is then very similar for tNAA and Tau, reinforcing the idea that Tau is neuronal in the cerebellum.

In the thalamus, the decrease in Tau content (*Figure 3B*) and the change in Tau diffusion properties (*Figure 4B*) point towards a change of compartmentation with age. Contrary to the trend in the cerebellum, $f_{sphere}$(Tau) consistently increases with age in the thalamus (*Figure 4—figure supplement 8*). This means that Tau diffuses in a space with fewer and fewer processes during development, which is incompatible with the hypothesis that Tau stays in the same compartment. Neurons, as well as microglia (*Cengiz et al., 2019*) and astrocytes (*Bushong et al., 2004*; *Somaiya et al., 2022*; *Clavreul et al., 2019*), develop with increasing ramification. Neurons are already branched at P10 in the thalamus (*Takeuchi et al., 2014*) (shown by a low $f_{sphere}$(tNAA, Glu) at P10). $f_{sphere}$(Tau) at P10 nears $f_{sphere}$(tNAA, Glu), indicating that Tau might be located in neuronal compartments in the neonatal thalamus. Although this hypothesis is not verifiable in existing literature, some in vitro studies show that Tau contributes to neuronal proliferation (*Hernández-Benítez et al., 2010*) and neuronal growth (*Mersman et al., 2020*). At P30, $f_{sphere}$(Tau) is closer to $f_{sphere}$(tCho, Ins), indicating that Tau might be located in glial compartments in adulthood. Indeed, immunohistochemistry studies locate Tau in adult thalamic glia (*Hussy et al., 2000*).

We suggest that Tau should be explored further as a cerebellar marker of early neuronal development, possibly more specifically of Purkinje cells. We confirm in this study the observation from *Tkác et al., 2003*, that both tCr and Tau have high spectroscopic signals in neonates. They are the two metabolites most reliably quantified up to P15 in both regions (continuing for Tau in the cerebellum up to P30), and the estimates of their diffusion properties are the most confident, contrary to the usual neuronal markers, tNAA and Glu. tCr stands as a good marker, too. However, there is no indication of a preferential neuronal compartmentation for this metabolite during early development; hence, it is less specific to neurons and probably even less to Purkinje cells. Because Tau seems to switch compartments in the thalamus between birth and adulthood, we do not think its diffusion properties reliably inform thalamic neuronal morphological development. In both human (*Hüppi et al., 1991*; *Pouwels et al., 1999*) and rhesus monkey (*Sturman et al., 1980*) brains, Tau content increases at birth and then decreases with age and is therefore a marker of the immature brain in both species. Although Tau concentration is low in the adult human brain, it is more concentrated in the cerebellum in adulthood (*Pouwels et al., 1999*; *Emir et al., 2012*). A similar finding was made in the adult rhesus monkey brain (*Sturman et al., 1980*), indicating a conserved pattern across species of Tau properties in the cerebellum. These observations form a good basis for translation. Although dMRS in infants has substantial challenges, they are worth trying to overcome for a putative cerebellar marker of early neuronal development which could help understand neurodevelopmental disorders (*Sathyanesan et al., 2019*).

We also observed a protracted increase in tNAA signal in the cerebellum compared to the thalamus. tNAA is usually described as a neuronal osmolyte, meaning it could reflect neuronal growth and maturation and be a marker of neuronal development. However, at the voxel level, its signal reflects the neuronal volumetric contribution, i.e., differences in neuronal densities might mask the effect of development. In contrast, we want to reiterate that dMRS is a normalised measure and dMRS outcomes are not affected by the neuronal density: they represent the average morphological properties of the cell types the metabolite belongs to.

We chose to interpret the dMRS data with two biophysical models: an analytical spheres and 'astrosticks' model and a computational morphometric model. The analytical model was applied simultaneously to the signal attenuation at high b-values and the ADCs at long diffusion times, and on individual animals. In addition to the valuable biophysical biomarkers, it provides a good signal representation to evaluate the significance of the differences between diffusion properties of individual datasets. The experimental design was not suitable for a randomly oriented infinite cylinders ('astrocylinders') model as in previous studies (*Palombo et al., 2017*; *Ligneul et al., 2019*). The mixing time used for the acquisition of signal decay at high b-values (TM = 100 ms) is too long to enable sensitivity to fibre radius (*Ligneul et al., 2024*), hence the choice of 'astrosticks' model. The sphere compartment was added to account for the high sphere fraction in the cerebellum, and for the nature of the neonatal brain (under-developed neurons). The root mean square errors (RMSEs) from the different models tested (*Figure 4—figure supplement 2*) tend to be more similar at P30, suggesting that it is less important to add a sphere compartment for modelling metabolites diffusion in the adult brain.

The computational morphometric model was applied to attempt to extract cell extension-related parameters, but the study design and data quality are suboptimal, and we could only reliably extract

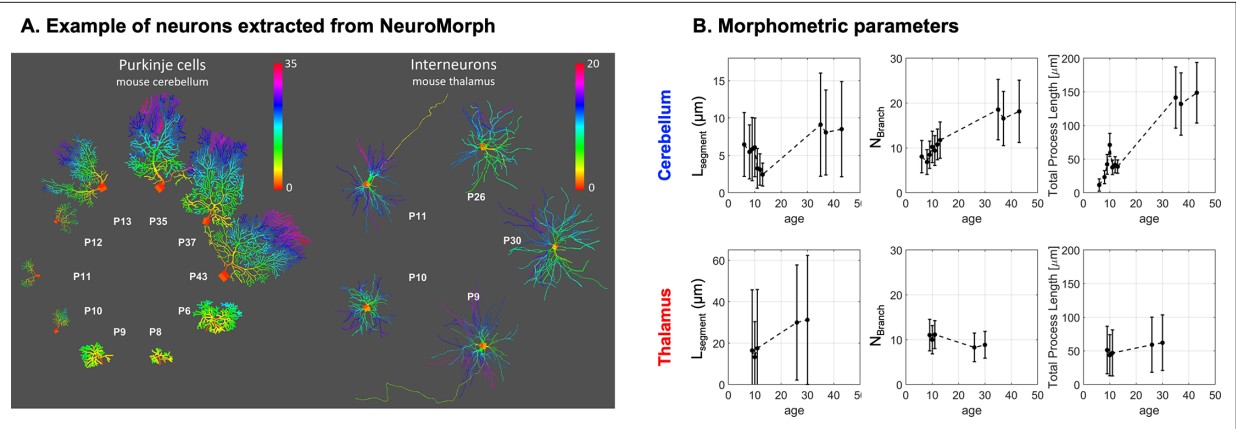

**Figure 6.** Morphometric parameters extracted from mouse real cells 3D reconstructions on NeuroMorph show a decrease of $L_{segment}$ up to P13 in developing cerebellar Purkinje cells, but not in developing thalamic interneurons. (**A**) Examples of cells extracted from NeuroMorph. The total number of cells used per age and region is given in **Supplementary file 3**. References: cerebellum P6-P10 (**Fukumitsu et al., 2016**), P10-P13 (**Jayabal et al., 2017**), P35-P43 (**Chen et al., 2013**), thalamus P9-P30 (**Hetsch et al., 2022**). (**B**) The values of branch length (Lsegment), branching order, and total process length are directly estimated from the morphology using the function stat_tree in the Trees MATLAB toolbox.

$L_{segment}$ and $D_{intra}$ (**Figure 4—figure supplement 6**). $D_{intra}$ is mostly stable with age in both regions. However, it sometimes drops or peaks, likely affecting the other parameters' estimations. The longest mixing time (TM = 1000 ms) is short compared to previous studies using a similar modelling approach (**Palombo et al., 2016**; **Ligneul et al., 2019**), meaning that the diffusion path is shorter (i.e. about 30 μm in this study compared to about 45 μm in previous studies, assuming $D_{intra}$ = 0.5 μm$^2$/ms). Note that the higher $D_{intra}$ of Tau is therefore an asset for probing early neuronal cerebellar growth. In our study, the number of embranchments ($N_{branch}$) was very poorly evaluated because of the high level of noise in the data. However, even with a better dataset quality and a more optimal study design, it is unlikely that the $N_{branch}$ estimation could reflect the Purkinje cell morphometry because of their size and complexity in adulthood. If we had robust DTI in all animals (**Figure 4—figure supplement 7**), we would have included the fibre dispersion in the modelling (notably for the cerebellum), as in **Ligneul et al., 2019**. However, since we acquired enough directions for the dMRS data to be powder-averaged, we expect that the WM contribution in the cerebellum is mostly reflected by a slight $f_{sphere}$ drop, as exemplified by the $f_{sphere,LT}$ drop when including WM (**Supplementary file 2**).

Finally, we expected that $L_{segment}$ estimated from the morphometric model in the cerebellum would increase regularly with age, matching the dendritic tree development. For neuronal, or partly neuronal metabolites, we report an $L_{segment}$ decrease first. Except for tNAA, which consistently decreases in the cerebellum with age, $L_{segment}$ decreases until P15 for tCr, Tau, and Glu and then increases, forming a U-shape curve. To better understand these results, we extracted the morphometries of developing Purkinje cells and developing thalamic interneurons (both in the mouse) from NeuroMorph.org. Interestingly, from these 3D reconstructed cells, we observe a decrease in $L_{segment, 3D-recon}$ up to P13 for Purkinje cells, whilst being quite stable in the thalamus (**Figure 6**), matching our non-invasive measurement trend. This could reflect the complexification of the dendritic tree following its expansion.

## Conclusions

This study proposes dMRS as a method to longitudinally and non-invasively measure cell-specific cerebellar and thalamic microstructural features during early development in rat neonates. We provide multiple lines of evidence, indicating the sensitivity of taurine to early neuronal cerebellar development, inviting further validation studies and opening interest for pathological conditions.

## Methods
### Animals

Experiments were approved by the United Kingdom Home Office under PPL P78348294. 18 Sprague-Dawley rats (9 females) were scanned at P5, P10, P15, P20, and P30. Pups came from 3 litters (6 pups

**Table 2.** Diffusion-weighted magnetic resonance spectroscopy (dMRS) acquisition parameters.

*dMRS acquisition parameters*

| Mixing time (ms) | 100 | | 500 | 750 | 1000 |
|---|---|---|---|---|---|
| Diffusion time (ms) | 104.25 | | 504.25 | 754.25 | 1004.25 |
| b-Values (ms/µm²) | 0.035, 3.035, 6, 10, 20, 30 | | 0.17, 3.17 | 0.26, 3.26 | 0.35, 3.35 |
| TR (ms) | 2000 | | 2500 | 2750 | 3000 |
| Total acquisition time | 19 min | | 5 min | 9 min | 19 min |
| Repetitions | 16x(4, 4, 6, 6, 8, 8) | | 16x(8,8) | 16x(10,10) | 16x(12,12) |
| Directions | 16 | | | | |
| TE | 34 ms (14 ms STE + 20 ms LASER) | | | | |
| Voxel size | | | | | |
| Age | P5 | P10 | P15 | P20 | P30 |
| Volume (µl) | 22.05 | 29.95 | 48.75 | 54.60 | 54.60 |

scanned per litter). Animals were induced with 4% isoflurane in oxygen and anaesthesia was maintained with 1–2% isoflurane in oxygen. Respiration was monitored with a breathing pillow. Temperature was monitored with a rectal probe, which was used as a cutaneous probe from P5 to P20 to avoid any risk of harm. Isoflurane level was adapted to keep the animals at 50–70 rpm and a warm water circulation heating pad was used to keep the animals at 36–37°C during scanning. Pups were gently held with foam and protected tape to minimise motion, because the neonatal skull is not suitable for the use of ear bars.

For logistic reasons, we could not acquire MEMRI for all litters. Animals from the second litter had manganese-enhanced anatomical scans. To this end, they were intraperitoneally injected with a 50 mg/kg solution of $MnCl_2$ 24 hr prior to the scan. Under the age of P10, the dam received the intraperitoneal injection, and the manganese was transferred to the neonates via the maternal milk (*Szulc et al., 2015*).

## MRI acquisitions

Data were acquired on a 7T Biospec Bruker MRI scanner (Paravision 360.1, max gradient strength: 660 mT/m), equipped with a mouse quadrature cryoprobe (4 channels, receive only), well suited for rat neonates, and a transmit-receive volume coil used for transmission. An anatomical scan around the region of interest (thalamus or cerebellum) to place the voxel (T2-weighted, 2D RARE, R=8, 0.2 mm isotropic resolution, TE = 33 ms, TR = 3500 ms). For six animals, this scan was replaced by a whole brain manganese-enhanced anatomical scan (T1-weighted, 3D MGE, TE = 2.4 s, TR = 62 ms, 0.15 mm isotropic resolution).

## dMRS acquisitions

A STELASER (*Ligneul et al., 2017*) sequence (TE = 34 ms, diffusion gradient duration, δ=5 ms) was used for dMRS acquisitions. The voxel was placed in the cerebellum for nine pups, and in the thalamus for the other nine pups. Voxel size was increased with age to follow brain growth. Diffusion-weighted spectra were acquired at four mixing times (TM) and two b-values (*Table 2* for detailed parameters): a small '$b_0$' used as a crusher for the stimulated echo, and $b_0$+3 ms/µm². Higher b-values were also acquired at the shortest diffusion time to probe restriction within cell bodies and enable inference of cell fraction and radius. The macromolecular baseline was acquired in the adult rodent with the same STELASER parameters (TE and δ), at the shortest TM = 100 ms, but the stimulated echo was preceded by a double-inversion recovery ($TI_1$=2000 ms, $TI_2$=710 ms). The b-value was set to 10 ms/µm² to reduce the residual contribution of bulk metabolites, and the tCr peak residual at 4 ppm (due to the shorter $T_1$) was corrected by subtracting a Lorentzian.

## Processing

### dMRS

Processing was achieved with a custom MATLAB (The Mathworks) routine (https://github.com/clemoune/dmrs-neonates/tree/main/A_dMRS_processing, preprocessed data can be found on https://zenodo.org/records/17107500):

- Signals from different coils were combined, using the water signal at the lowest b-value and shortest diffusion time.
- Outliers were identified and discarded, as described in *Ligneul et al., 2023*.
  - For each transient i, the mean absolute upfield signal (range 1.9–4.5 ppm), $M_{up}(i)$ was computed.
  - Transients affected by motion are characterised by a clear drop of $M_{up}(i)$. Transients falling below the median $(M_{up})$ – 3xMAD threshold were discarded.
  - Experiments at high b-values are more susceptible to changing gradient direction (particularly if the voxel contains white matter tracts). To predict its effect across b-values and overcome the small number of acquisitions per unique condition (i.e. a (b-value, direction) pair), a diffusion kurtosis tensor was computed based on $M_{up}$. The MAD calculated at b=0 was used as a 'ground truth' MAD and was reported to higher b-values (per direction). High b-value acquisitions are more easily corrupted by motion and therefore present an artificially high MAD.
- Individual transients were corrected for phase and frequency drifts and averaged across all diffusion directions (powder-averaged) via an arithmetic mean. In general, the metabolites signal was visually detectable on single transients (with a line broadening of 5 Hz), except at b=20,30 ms/µm$^2$ at P5, rendering these acquisitions more susceptible to motion artefacts.
- Any water residual was removed using a singular value decomposition method centred around the water region. If the baseline was heavily distorted despite this step due to very poor water suppression, the acquisition was discarded from the analysis.

Spectra were subsequently fitted with LCModel. The simulated basis set contained the following metabolites: Ace, Ala, Asp, Cr, GABA, Gln, Glu, Gly, GPC, GSH, Ins, sIns, Lac, NAA, NAAG, PCho, PCr, Tau, PE, and included a macromolecule (MM) baseline acquired experimentally (double-inversion recovery and b=10 ms/µm$^2$).

Diffusion properties of NAA + NAAG (tNAA), Glu, Cr + PCr (tCr), Tau, GPC + PCho (tCho), and Ins are reported. In order to model both signal compartmentalisation and restricted diffusion by cell membranes, we investigated metabolites diffusion through two sets of measurements: (i) the direction-averaged signal decay as a function of the diffusion weighting b, keeping the diffusion time constant at 104.25 ms (time between diffusion gradients Δ fixed at 105.91 ms and δ fixed at 5 ms) and varying the strength of the diffusion-sensitising gradients; and (ii) the ADC as a function of increasing diffusion time (keeping δ fixed at 5 ms). For the measurements (i), data are reported as the ratio S(b)/S(b=0). For the measurements (ii), ADCs diffusion time dependencies are reported. ADCs are calculated following $ADC = \frac{-1}{b-b_0}\log\frac{S(b)}{S(b_0)}$. Negative ADCs were systematically removed as they point to a quantification error.

In the cerebellum, metabolites' signal attenuation at P5 is always much stronger and quite far off the other time points. The macromolecules' signal attenuation at high b-values is supposed to remain low (*Ligneul et al., 2017*) and can be used as an internal probe for motion artefact (*Ligneul and Valette, 2017*). However, the macromolecules' signal attenuation is quite strong at P5 in the cerebellum (*Figure 4—figure supplement 9*), pointing out to a residual motion artefact despite careful processing. The spectroscopic signal in individual transients was too low at high b-values, notably in the cerebellum: it is possible that the spectral realignment failed.

### MEMRI

The three first echoes of each MGE acquisition were averaged and denoised using the DenoiseImage function from the MINC toolbox, based on *Coupe et al., 2008*. For each time point, an image with no obvious artefact and a good visual CNR was taken as the initial model and transformed into the Fischer atlas (*Goerzen et al., 2020*) space. Images were then processed using registration_tamarack.py with the MAGeT segmentation in the available Pydpiper pipelines (*Friedel et al., 2014*). Due to the large deformations between P30 and P5, the P30 time point could not be used as a reference. An adequate atlas

from P15 was generated in a couple of iterations (using a P20 atlas as intermediary). It was then used as a reference atlas and reference time point for the pipeline. The olfactory bulb and the more frontal parts were poorly registered because the image often had an intensity drop there, but it did not affect the registration of caudal regions, such as the cerebellum and the thalamus. Images with strong intensity differences, or corrupted by motion, were removed from the analysis. The registration did not work well at P5, even when attempting to use an atlas generated at P10 (that is already slightly distorted).

## Analysis

Signal decays at high b-values (at TM =100 ms) and ADCs at varying diffusion times were fitted together with a spheres and 'astrosticks' (corresponding to randomly oriented sticks) model using the functions implemented in DIVE (https://git.fmrib.ox.ac.uk/fsl/DIVE). The DIVE toolbox uses the Gaussian phase distribution approximation. We fitted the signal to:

$$\frac{S}{S_0} = f_{sphere} * S_{sphere}\left(D_{intra}, R_{sphere} \mid b_{value}, t_d, d\right) + \left(1 - f_{sphere}\right) * S_{sticks}\left(D_{intra} \mid b_{value}\right)$$

Free parameters were R (sphere radius) and $f_{sphere}$ (sphere fraction). $D_{intra}$ (i.e. the metabolite intracellular free diffusivity) was fixed for both compartments at 0.5 µm$^2$/ms. Previous studies measuring brain metabolites diffusion properties at ultrashort diffusion times (*Ligneul and Valette, 2017*; *Marchadour et al., 2012*) estimated the intracellular free diffusivity at 0.5–0.6 µm$^2$/ms for most metabolites (except Tau). However, other studies using different (less direct) approaches reported lower $D_{intra}$ (*Palombo et al., 2016*; *Palombo et al., 2017*); hence, we tested a range of reasonable $D_{intra}$. $D_{intra}$ was fixed at 0.3, 0.4, 0.5, 0.6, 0.7, and 0.8 µm$^2$/ms. RMSEs between model fit and data were equivalent for all $D_{intra}$, except for Tau in the thalamus where increasing $D_{intra}$ improved the fit, particularly until $D_{intra}$ = 0.5 µm$^2$/ms. Moreover, parameter estimation was unrealistic for low $D_{intra}$. Increasing $D_{intra}$ between 0.5 µm$^2$/ms and 0.7 µm$^2$/ms did not change the trend of the results, except for Tau in the thalamus (see *Figure 4—figure supplement 2* and Discussion). The fit was made robust to missing data (when excluded during processing).

To test for the model accounting best for our data, we tested different analytical models. RMSE between model fit and data was calculated for 'astrosticks' only (1 free parameter, $D_{intra}$), 'astrocylinders' (corresponding to randomly oriented infinite cylinders, 2 free parameters: $D_{intra}$ and cylinder radius), spheres and astrosticks without constraint (3 free parameters) and spheres and astrosticks with $D_{intra}$ fixed at 0.5 µm$^2$/ms (*Figure 4—figure supplement 2*). RMSE was the lowest for spheres and astrosticks and was very similar with and without the $D_{intra}$ constraint. Hence, the model with the least free parameters was chosen. Moreover, the results from the constrained spheres and astrosticks model were more stable and physically meaningful with the constraint on $D_{intra}$. The code and data used for the analysis can be found on: https://github.com/clemoune/dmrs-neonates/tree/main/B_Astrosticks_Spheres_model.

## Long diffusion times

Average ADCs at varying diffusion times were fitted with the morphometric model introduced in *Palombo et al., 2016*. Modelling was performed using the dictionary generated for *Ligneul et al., 2019*, with TM = 100, 750 ms predicted by interpolation, and a random forest regressor ('treeBagger' function in MatlabR2020b, with tree depth = 20; number of trees = 200, and other model hyper-parameters as per default) trained in a fully supervised fashion, using the ADCs time dependencies as input and the ground truth morphological parameters as targets; as a loss function, the mean squared error between predicted and ground truth parameter values was used. Parameters fitted were $D_{intra}$, $N_{branch}$, $L_{segment}$, $SDN_{branch}$, and $SDL_{segment}$. Boundaries for fitting the model to data were respectively [0.1, 1], [2, 25], [5, 100], [2, 3], and [5, 10]. Given the data noise, only $L_{segment}$ and $D_{intra}$ are estimated robustly. Fits for all parameters are shown in Supplementary Information. The code and data used for the analysis can be found on: https://github.com/clemoune/dmrs-neonates/tree/main/C_Morphometric_Modelling.

## Statistical analyses

Statistical analyses were run with R, using the 'lmerTest' (*Kuznetsova et al., 2017*) package. To assess whether diffusion properties were significantly different in the two regions during development, we

applied a linear mixed effect model to the parameters extracted from the spheres + 'astrosticks' model (sphere fraction and sphere radius), using age and region as fixed effects, and a random intercept for pups ID. An analysis of variance enabled us to identify whether the inclusion of an interaction term age*region was necessary or not. Results were corrected for multiple hypotheses testing with a Bonferroni correction (6 metabolites, 2 parameters = 12 hypotheses). The threshold corrected p-value is $4.2e^{-3}$ for $\alpha=0.05$ and $8.3e^{-4}$ for $\alpha=0.01$.

## Acknowledgements

This work was funded by Wellcome. The Wellcome Centre for Integrative Neuroimaging is supported by core funding from the Wellcome Trust (203139/Z/16/Z and 203139/A/16/Z). For the purpose of Open Access, the corresponding author has applied a CC BY public copyright licence to any Author Accepted Manuscript version arising from this submission. MP is supported by UKRI Future Leaders Fellowship (MR/T020296/2).

## Additional information

### Competing interests

Saad Jbabdi, Jason P Lerch: Reviewing editor, eLife. The other authors declare that no competing interests exist.

### Funding

| Funder | Grant reference number | Author |
|---|---|---|
| Wellcome Trust | 203139/Z/16/Z | Jason P Lerch |
| Wellcome Trust | 203139/A/16/Z | Jason P Lerch |
| UK Research and Innovation | MR/T020296/2 | Marco Palombo |

The funders had no role in study design, data collection and interpretation, or the decision to submit the work for publication. For the purpose of Open Access, the authors have applied a CC BY public copyright license to any Author Accepted Manuscript version arising from this submission.

### Author contributions

Clémence Ligneul, Conceptualization, Data curation, Software, Formal analysis, Supervision, Funding acquisition, Investigation, Methodology, Writing – original draft, Project administration, Writing – review and editing; Lily Qiu, Investigation; William T Clarke, Methodology, Writing – review and editing; Saad Jbabdi, Software, Methodology, Writing – review and editing; Marco Palombo, Conceptualization, Software, Methodology, Writing – review and editing; Jason P Lerch, Conceptualization, Resources, Writing – review and editing

### Author ORCIDs

Clémence Ligneul https://orcid.org/0000-0001-5673-3009
William T Clarke https://orcid.org/0000-0001-7159-7025
Saad Jbabdi https://orcid.org/0000-0003-3234-5639

### Ethics

All experiments were approved by the United Kingdom Home Office, and realised under the PPL number P78348294.

Reviewer #1 (Public review): https://doi.org/10.7554/eLife.96625.4.sa1
Author response https://doi.org/10.7554/eLife.96625.4.sa2

# Additional files

## Supplementary files

Supplementary file 1. Mean relative CRLB (Cramér-Rao lower bound) for tNAA, Glu, tCr, Tau, tCho, and Ins for all diffusion conditions in both regions.

Supplementary file 2. Estimation of $f_{sphere,LT}$ derived using the ratio of the relative thicknesses of the EGL, IGL, PL ('cell body-like' layers), and ML ('process-like' layer). 'with WM' means that the white matter layer was included in the 'process-like' layers (in addition to ML). Data come from manual measurements of literature cerebellar figures. References are at the bottom of the Supplementary Information.

Supplementary file 3. Number of 3D neuron reconstructions used from Neuromorph. Supplementary information to *Figure 6*.

MDAR checklist

## Data availability

The code related to preprocessing and modelling is shared on GitHub: https://github.com/clemoune/dmrs-neonates/ (copy archived at *clemoune, 2025*). Each GitHub subdirectory contains the processed data necessary for the subsequent analysis, except for the preprocessed dMRS data hosted on Zenodo. The preprocessed dMRS data are shared on Zenodo: https://doi.org/10.5281/zenodo.17107499.

The following dataset was generated:

| Author(s) | Year | Dataset title | Dataset URL | Database and Identifier |
|---|---|---|---|---|
| Clémence L | 2025 | Preprocessed dMRS data (thalamic and cerebellar development) | https://doi.org/10.5281/zenodo.17107499 | Zenodo, 10.5281/zenodo.17107499 |

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
