## [Editor Report · eLife Assessment]

This study presents a **valuable** investigation into cell-specific microstructural development in the neonatal rat brain using diffusion-weighted magnetic resonance spectroscopy. The evidence supporting the core claims is **solid**, with innovative in vivo data acquisition and modeling, noting residual caveats with regard to the limitations of diffusion-weighted magnetic resonance spectroscopy for strict validation of cell-type-specific metabolite compartmentation. In addition, the study provides community resources that will benefit researchers in this field. The work will be of interest to researchers studying brain development and biophysical imaging methods.

---

## [Referee Report · Reviewer #1 (Public review)]

In this work, Ligneul and coauthors implemented diffusion-weighted MRS in young rats to follow longitudinally and in vivo the microstructural changes occurring during brain development. Diffusion-weighted MRS is here instrumental in assessing microstructure in a cell-specific manner, as opposed to the claimed gold-standard (manganese-enhanced MRI) that can only probe changes in brain volume. Differential microstructure and complexification of the cerebellum and the thalamus during rat brain development were observed non-invasively. In particular, lower metabolite ADC with increasing age were measured in both brain regions, reflecting increasing cellular restriction with brain maturation. Higher sphere (representing cell bodies) fraction for neuronal metabolites (total NAA, glutamate) and total creatine and taurine in the cerebellum compared to the thalamus were estimated, reflecting the unique structure of the cerebellar granular layer with a high density of cell bodies. Decreasing sphere fraction with age was observed in the cerebellum, reflecting the development of the dendritic tree of Purkinje cells and Bergmann glia. From morphometric analyses, the authors could probe non-monotonic branching evolution in the cerebellum, matching 3D representations of Purkinje cells expansion and complexification with age. Finally, the authors highlighted taurine as a potential new marker of cerebellar development.

From a technical standpoint, this work clearly demonstrates the potential of diffusion-weighted MRS at probing microstructure changes of the developing brain non-invasively, paving the way for its application in pathological cases. Ligneul and coauthors also show that diffusion-weighted MRS acquisitions in neonates are feasible, despite the known technical challenges of such measurements, even in adult rats. They also provide all necessary resources to reproduce and build upon their work, which is highly valuable for the community.

From a biological standpoint, claims are well supported by the microstructure parameters derived from advanced biophysical modelling of the diffusion MRS data.

Specific strengths:

(1) The interpretation of dMRS data in terms of cell-specific microstructure through advanced biophysical modelling (e.g. the sphere fraction, modelling the fraction of cell bodies versus neuronal or astrocytic processes) is a strong asset of the study, going beyond the more commonly used signal representation metrics such as the apparent diffusion coefficient, which lacks specificity to biological phenomena.

(2) The fairly good data quality despite the complexity of the experimental framework should be praised: diffusion-weighted MRS was acquired in two brain regions (although not in the same animals) and longitudinally, in neonates, including data at high b-values and multiple diffusion times, which altogether constitutes a large-scale dataset of high value for the diffusion-weighted MRS community.

(3) The authors have shared publicly data and codes used for processing and fitting, which will allow one to reproduce or extend the scope of this work to disease populations, and which goes in line with the current effort of the MR(S) community for data sharing.

Specific weaknesses:

Ligneul and coauthors have convincingly addressed and included my comments from the first and second round in their revised manuscript.

I believe the following conceptual concerns, which are inherent to the nature of the study and do not require further adjustments of the manuscript, remain:

(1) Metabolite compartmentation in one cell type or the other has often been challenged and is currently impossible to validate in vivo. Here, Ligneul and coauthors did not use this assumption a priori and supported their claims also with non-MR literature (eg. for Taurine), but the interpretation of results in that direction should be made with care.

(2) Longitudinal MR studies of the developing brain make it difficult to extract parameters with an "absolute" meaning. Indirect assumptions used to derive such parameters may change with age and become confounding factors (brain structure, cell distribution, concentrations normalizing metabolites (here macromolecules), relaxation times...). While findings of the manuscript are convincing and supported with literature, the true underlying nature of such changes might be difficult to access.

(3) Diffusion MRI in addition to diffusion MRS would have been complementary and beneficial to validate some of the signal contributions, but was unfeasible in the time constraints of experiments on young animals.

---

## [Author Response]

The following is the authors’ response to the previous reviews

We thank the reviewers once again for their careful evaluation of the revised manuscript and for their constructive suggestions. In response to the remaining recommendations, we have made minor amendments to the manuscript. The main changes are as follows:

• Metabolite Concentrations: we now report them more conventionally, i.e. normalised by water content. The original normalisation by the absolute MM content has been retained in the supplementary information, as MMs are an endogenous tissue probe (i.e., not dependent on cerebrospinal fluid). The fact that both water and MM normalisation provide similar trends supports the robustness of our conclusions. We have also updated Figure S2 to include the absolute MM concentrations, raw water content, and the MM-to-water ratios for each time point.

• Taurine Interpretation: We have revised the wording related to the interpretation of taurine findings to clarify that we present a set of converging observations suggesting taurine may serve as a marker of early cerebellar neurodevelopment, rather than asserting it as a definitive conclusion.

**Comments to the editor & reviewers:**

We sincerely thank the reviewers and the editor for their valuable feedback, which has significantly improved the manuscript since its initial submission.

Please note a correction in Figure S2 (added during the previous revision round): the reported evolution of metabolite/water concentrations has changed due to an earlier error in calculating the water peak integral, which has now been corrected.

While we recognise that a study and manuscript can always be improved, we prefer not to make further changes at this stage. We cannot conduct new experiments, and redesigning the model falls outside the scope of this work. Additionally, we believe that further altering the manuscript’s structure could lead to unnecessary confusion rather than clarity.